# Emerging Advances of Detection Strategies for Tumor-Derived Exosomes

**DOI:** 10.3390/ijms23020868

**Published:** 2022-01-14

**Authors:** Huijuan Cheng, Qian Yang, Rongrong Wang, Ruhua Luo, Shanshan Zhu, Minhui Li, Wenqi Li, Cheng Chen, Yuqing Zou, Zhihua Huang, Tian Xie, Shuling Wang, Honghua Zhang, Qingchang Tian

**Affiliations:** 1College of Pharmacy, Hangzhou Normal University, Hangzhou 311121, China; huijuancheng1994@163.com (H.C.); qiandecade@163.com (Q.Y.); rong123450609@163.com (R.W.); 18327602598@163.com (R.L.); lmh5546@163.com (M.L.); l17816111674@163.com (W.L.); jinshaner1@dingtalk.com (C.C.); 2021112012252@stu.hznu.edu.cn (Y.Z.); hzh9799@gmail.com (Z.H.); xbs@hznu.edu.cn (T.X.); 2Key Laboratory of Elemene Class Anti-Cancer Chinese Medicines, Hangzhou Normal University, Hangzhou 311121, China; 3Engineering Laboratory of Development and Application of Traditional Chinese Medicines, Hangzhou Normal University, Hangzhou 311121, China; 4Collaborative Innovation Center of Traditional Chinese Medicines of Zhejiang Province, Hangzhou Normal University, Hangzhou 311121, China; 5Public Health Institutes, Hangzhou Normal University, Hangzhou 311121, China; 2019213001043@stu.hznu.edu.cn

**Keywords:** exosomes, tumor-derived exosomes, molecular biomarkers, detection methods

## Abstract

Exosomes derived from tumor cells contain various molecular components, such as proteins, RNA, DNA, lipids, and carbohydrates. These components play a crucial role in all stages of tumorigenesis and development. Moreover, they reflect the physiological and pathological status of parental tumor cells. Recently, tumor-derived exosomes have become popular biomarkers for non-invasive liquid biopsy and the diagnosis of numerous cancers. The interdisciplinary significance of exosomes research has also attracted growing enthusiasm. However, the intrinsic nature of tumor-derived exosomes requires advanced methods to detect and evaluate the complex biofluid. This review analyzes the relationship between exosomes and tumors. It also summarizes the exosomal biological origin, composition, and application of molecular markers in clinical cancer diagnosis. Remarkably, this paper constitutes a comprehensive summary of the innovative research on numerous detection strategies for tumor-derived exosomes with the intent of providing a theoretical basis and reference for early diagnosis and clinical treatment of cancer.

## 1. Introduction

The currently defined exosome (40–150 nm) was discovered in sheep reticulocytes in 1983 [1,2]. Johnstone et al. [3] tracked transferrin receptors in the process of reticulocytes’ maturation and suggested that the generation of exosomes is the mechanism for the loss of transferrin receptors in mature erythrocytes. More and more studies have shown that exosomes exist in bodily fluids and participate in intercellular communication. Furthermore, they appear as specific early warnings for cancer such as breast cancer [4] and pancreatic cancer [5], tumors such as melanoma [6], and mental illness such as Alzheimer’s [7] and Parkinson’s disease [8,9].

Tumors possess complex tissues composed of matrix components (such as fibroblasts, mesenchymal cells, smooth muscle cells, pericytes, etc.), immune cells, platelets, and endothelial cells related to heterogeneous subcells. Through the secretion of soluble factors, cytokines, and exosomes, cancer cells continuously reshape their environment by recruiting and activating surrounding cells [10]. Exosomes are secreted from cancer cells, and the specific antigens on their surface can reflect the nature of donor cells [11]. A multifold of the current reviews in exosomes focused on exosomal physical characterization, separation, and labeling for semiquantitative analyses [12,13,14,15,16]. However, introducing novel detection and clinical diagnostic methods of tumor-derived exosomes is not comprehensive enough. This paper summarizes the biological origin, composition, molecular markers, and clinical diagnosis of exosomes, analyzes the relationship between exosomes and tumors, and summarizes the precise and quantitative detection methods of tumor-derived exosomes and their clinical potential.

## 2. Exosomes

### 2.1. Synthesis and Release of Exosomes

Cells release various types of vesicles to transmit information between cells. These vesicles are divided into apoptotic bodies (>1000 nm), microvesicles (200~1000 nm), and exosomes (40~150 nm) [16,17]. The latter are membranous vesicles formed by living cells through the process of “endocytosis-fusion-efflux”, divided into the following four stages: initiation, endocytosis, multivesicular bodies (MVB) formation, and exosomal secretion [18]. Nano-scale exosomes penetrate cell membranes and infiltrate many biological fluids such as blood, urine, saliva, synovial fluid, breast milk, etc. The schematic diagram of the synthesis and release of exosomes is displayed in Figure 1. The early stage of exosomal formation consists of intraluminal vesicles produced by cells through endocytosis. With the entry of heat shock proteins, microRNAs (miRNAs), enzymes, and other loading molecules in the cytoplasm, more and more intramural vesicles are formed. Finally, MVBs are formed, which subsequently fuse with the cell membrane to secrete exosomes outside the cell [18]. The modulation of MVB, exosomal synthesis and release is through the endosomal sorting complexes required for the transport (ESCRT) pathway or an ESCRT-independent mechanism during the process [19,20,21]. MVBs are eventually either delivered to the lysosome to be degraded along with its components or fused with the cell’s plasma membrane to release its content, including exosomes, into the extracellular space [22,23,24,25]. Although the effect of ESCRT in MVB synthesis has been extensively studied, whether MVB produced by the ESCRT-dependent pathway is degraded by lysosomes or fused with the cytoplasm to release exosomes, as well as its regulatory mechanism, remains elusive [26,27,28].

### 2.2. Components in Exosomes

#### 2.2.1. Exosomal Nucleic Acid

Specific nucleic acids that mutant mRNA such as the heterogeneous nuclear ribonucleoproteins (hnRNPH1) in serum contained in exosomes may possibly serve as reliable biomarkers for liver cancer [29]. Unique miRNAs in exosomes could also message diagnosis and observe the evolution of cancer. For example, exosomal miR-165-3p is increased in the urine of patients with bladder cancer and relates to advanced disease stage [30]. In another instance, exosomal miR-423-5p level in the serum of patients with gastric carcinoma enhance gastric cancer cell proliferation and migration by targeting the tumor suppressor of fused protein (SUFU) [31]. Analyses of exosomes derived from plasma revealed a miRNA profile that may enable the detection of breast tumors [32].

DNA in exosomes may provide information about cancer-specific mutations [33,34]. Indeed, genomic sequencing revealed that exosomes in the serum of glioma cancer patients carry human-specific DNA sequences relevant for glioblastoma multiforme (GBM) biology [35]. Additionally, driver mutations associated with pancreatic ductal adenocarcinoma (PDAC) were identified in this exosomal DNA [33]. The chances of whole genomic sequencing of serous exosomes in cancer patients may provide information for diagnosing and predicting treatment outcomes.

#### 2.2.2. Exosomal Proteins

The proteins in exosomes are composed of the transmembrane family and endosomal proteins. On the one hand, the transmembrane family includes membrane transport protein (Annexins and Rabs), regular immune molecules such as major histocompatibility complexes (MHC), class I, and MHC class II molecules; acceptors, integrins, transmembrane proteins (CD81, CD82, CD9, and CD63), etc. On the other hand, endosomal proteins include chaperone proteins (Hsp60, Hsp70, Hsc70, and Hsp90), skeleton proteins (Profilin, Tubulin, Cofilin, etc.), synthetic proteins (Clatherin, Ubiqulin, Alix and TSG-101), signal proteins such as ADP-ribosylation factors 1 (ARF1), phosphatidylinositol 3-kinase (PI3K), and cell division cycle 42 (CDC42), enzymes such as pyruvate kinase (PK), protein kinase G (PKG), ATPases, etc., and cancer-associated specific markers such as glypican-1 (GPC-1), epithelial cell adhesion molecule (EpCAM), programmed death ligand 1 (PD-L1), and epidermal growth factor receptor (EGFR). Exosomal proteins likely mirror their cellular origin and may also facilitate the detection of cancer. As a matter of fact, GPC-1 was detected in the serum of pancreatic carcinoma, which discriminated patients with pancreatic ductal adenocarcinoma from those with benign pancreas diseases, such as chronic pancreatitis [36]. Moreover, exosomes containing CD63 significantly increase in melanoma patients and can be served as markers for tumor detection [37,38].

Exosomes not only play a vital role in regulating conventional physiological processes, such as stem cell maintenance [39], tissue repair [40], and immune regulation [41], but also play crucial roles in tumor treatment [42]. The schematic diagram of exosomal components is illustrated in Figure 2.

Exosomal molecular markers have brought innovative ideas to the diagnosis and treatment of tumors, indicating satisfactory clinical significance. The exosome is a tremendous hub in physiological or pathological processes; therefore, the development of non-invasive diagnostic methods with higher sensitivity and specificity based on exosomes has enormous potential for the early diagnosis, treatment evaluation, and prognostic analysis of diseases. Now, the relationship between tumor and exosomal molecular markers is summarized below (Table 1).

## 3. Exosomes and Tumor Development

The cargo of extracellular vesicles reflecting the cell of origin has opened a new frontier for non-invasive biomarker discovery in oncology [79]. Specific tumor-derived extracellular vesicles are released into the extracellular space and can be found in the plasma, serum, and urine of patients presenting with tumors [80]. The main functions of exosomes in intercellular communication are to promote the growth of primary cancers, stimulate angiogenesis, activate stromal fibroblasts, remodel cancer extracellular matrix, and inhibit the host’s immune response [10]. In the process of tumor formation, exosomes can use advantages of their structure and composition to modulate immune system function, form a microenvironment that is conducive to tumor formation, and induce malignant tumor behavior [81]. Sung et al. [82] suggested that the auto-secretion of exosomes boosted the movement of cancer cells by enhancing the transient polarization and adhesion in other ways. It is reasonable to hypothesize that the biological occurrence of inhibitory exosomes may largely lead to the loss of cancer cell migration, which will provide a feasible strategy for eliminating exosomes secreted in the tumor area and inhibiting the occurrence, development, and metastasis of cancer cells. Lopatina et al. [83] demonstrated how the IL-3Rα blockade on tumor-endothelial cells reprograms extracellular vesicles, which then acquire the ability to change the expression of Vimentin, β-catenin, and TWIST1, and reduce angiogenesis and the metastatic spread of primary tumors. By comparing the miRNA map of ovarian cancer with the tumor-derived exosomal map separated from the oophoroma patient, Taylor et al. [84] demonstrated the association between miRNA and circulating tumor-derived exosomes, and the results implied that miRNA analysis of circulating tumor exosomes could be used as an alternative diagnostic biomarker for biopsy analysis, extending its usefulness to screening asymptomatic populations.

The relationship between exosomes and tumor development is summarized below: tumor-derived exosomes transfer carcinogens such as oncoproteins, mRNA, etc., from aggressive tumor cells to resting or normal cells, accelerating the malignant transformation of these cells, thereby bolstering the occurrence of tumors. These exosomes can regulate the occurrence of tumors by modulating the immune system, reshaping the tumor microenvironment, and promoting angiogenesis. The schematic diagram of the relationship between tumor-derived exosomes and tumors is presented in Figure 3.

## 4. Novel Detection Methods of Tumor-Derived Exosomes

The size and density of exosomes are relatively close to lipoproteins and protein complexes, and diverse separation and purification methods will affect their purity and quality. At present, a variety of separation and purification schemes have been established based on exosomes’ physical and chemical properties, such as ultracentrifugation, immunomagnetic beads, commercial kits, etc. [85,86]. In contrast, exosomes with protein or nucleic acid biomarkers are detected by enzyme-linked immunosorbent assay (ELISA), polymerase chain reaction (PCR), DNA sequencing, or microarray analysis [87]. In order to further strengthen the feasibility of the application of exosomal molecular biomarkers, researchers have been struggling to optimize the procedures for exosomal separation. Due to the benefits of speed, integration, and small sample size, microfluidic strategies have attracted more and more attention [88,89]. At the same time, there are also emerging interdisciplinary technologies based on fluid mechanics and other different engineering fields (such as materials, microelectronics, biomedical engineering, etc.) for the identification of exosomes [90]. Therefore, this article reviews the novel detection methods of tumor-derived exosomes in combination with exosomal molecular biomarkers and tumor diagnosis, which provides a theoretical basis and reference for early cancer diagnosis and clinical treatment.

Because a single method fitting a variety of sample sources is not feasible, efforts have been made to exploit different physicochemical and biochemical properties of exosomes. To date, seven classes of exosome detection strategies have been reported, including size-exclusion chromatography, droplet digital ELISA (ddELISA), digital droplet PCR (ddPCR), microfluidic techniques, surface-enhanced Raman scattering (SERS) technology, aptamer-based separation method, quantum dot-based exosome quantification, with a unique sets of pros and cons for each technique (Table 2).

### 4.1. Separation and Purification

Separation and purification are the first steps for detection. Size exclusion chromatography is a chromatographic method that separates molecules in a solution according to their size or molecular weight [108,109]. The difference between the pore diameter of the gel and the molecular size of the sample is used for separation. The research separated the test substance to an agarose gel chromatography column, further separated the mixed components of the test substance according to the difference in molecular size, and successfully split extracellular vesicles from high-density lipoprotein (HDL) and protein step by step [110]. This provides a groundbreaking idea for the separation of nanoscale extracellular vesicles in a large mass. Zhang et al. [111] developed a method to efficiently separate extracellular vesicles from HDL, low-density lipoprotein (LDL), and very-low-density lipoprotein (VLDL) based on differences in surface charge and molecular size by agarose gel electrophoresis. This method is of immense value for separating high-purity extracellular vesicles from plasma and other lipoprotein-contaminated samples.

### 4.2. Digital Detection of Proteins and Nucleic Acids

The measurements of multiple biomolecules in the same biological sample are vital for diagnosing or classifying clinical diseases. However, the content of these disease-related biomarkers in biological fluids is usually very low, so ultra-sensitive measurement methods are required [112]. In recent years, digital ELISA and digital PCR platforms have revolutionized detection technologies for the absolute quantification of nucleic acids and proteins with microfluidics [113,114,115]. Cohen et al. [116] developed a ddELISA for ultrasensitive detection of single protein molecules. This is a quantitative method to distinguish low-level proteins and exosomes with digital ELISA and droplet microfluidics. The ddELISA achieves maximum sensitivity by improving sampling efficiency and counting more target molecules. This will facilitate the discovery of countless biomarkers that have never been measured before clinical applications. The ddPCR is an emerging technology capable of absolute nucleic acid quantification without using standard curves [117]. This ddPCR technology combines microfluidics and proprietary surfactant chemistries to separate PCR samples into water-in-oil droplets [98]. Meanwhile, Cho et al. [118] reported the identification of *Mycobacterium tuberculosis* DNA from exosomes isolated from tuberculosis patients and suggested higher sensitivity by using exosomal DNA than using total DNA. Additionally, tumor-derived exosomes can be identified by tumor-specific biomarkers via antibody labeling. Lin et al. [119] developed a dual-target-specific aptamer recognition activated in situ connection system on exosome membrane combined with ddPCR for the quantitation of tumor-derived exosomal PD-L1 (Exo-PD-L1). The working principle of the highly sensitive quantification of tumor-derived Exo-PD-L1 using aptamer-based proximity ligation assay is as follows: Two types of designed aptamer probes against EpCAM and PD-L1 were utilized to simultaneously label both types of protein biomarkers on the exosome. The extended parts of two aptamers are in close proximity because of the fluidity of the exosomal membrane. After ligation, ddPCR is performed to quantify the ligation products. Similarly, Wang et al. [113] developed a ddPCR system based on an oil saturated polydimethylsiloxane (PDMS) microfluidic chip platform for the quantification of lung cancer related miRNAs. This droplet PCR system provides new possibilities for highly sensitive and efficient detection of cancer-related genes.

### 4.3. Microfluidic Technology

Recently, microfluidic platforms have revolutionized detection technologies with high-throughput and rapid detection of exosomes, which have been employed in subpopulation typing [100,101,120,121]. Microfluidic technology has the upper edge in terms of low material consumption, low pollution, and a small sample required for analysis, and hence is considered a promising technology [122].

#### 4.3.1. Acoustics-Based Microfluidics

The acoustics-based microfluidic device is a robust particle manipulation strategy to exert radiation forces on particles. Under acoustic pressure, particles endure differential forces according to their mechanical properties (size, density, compressibility) [123,124]. Lee et al. [125] designed a microfluidic chip based on acoustic fluid technology to separate microvesicles and exosomes from the culture supernatant of ovarian cancer cells. Interactive digital converting electrodes were installed in the middle of the device to control the radiating force of the acoustic wave and adjust the sample flow rate. Large particles moved more rapidly along both sides of the flow, whereas smaller particles flowed along the middle. At the end of the device, the large and small particles were collected by different channels. This system could further extend the usefulness of acoustofluidics for microvesicle analysis. Indeed, Wu et al. [122] reported that the purification efficiency of exosomes and other extracellular vesicles from whole blood reached 98.4% by integrating acoustic and microfluidic technologies. This integrated chip technology can directly isolate exosomes and other types of extracellular vesicles from undiluted whole blood samples in an automated manner.

#### 4.3.2. Immunoaffinity-Based Microfluidics

The capturing methods combined with immune-affinity and microfluidic technology are highly efficient, specific, and rapid, and are currently mainly applied for tumor-derived exosomal separation. To realize the digital qualification of target exosomes, a droplet digital microfluidic chip was fabricated for single exosome counting to identify the heterogeneity of exosomes. Liu et al. [126] described a new digital detection method based on droplet microfluidics technology to count single exosomes for breast cancer diagnosis. First, the immuno-magnetic beads are used to capture the exosomes in the sample, based on the Poisson distribution principle. Then, an ExoELISA reaction is carried out to connect the captured exosomes with a fluorescent reaction enzyme. Next, the magnetic bead complex is formed on the microdroplet chip, together with the fluorescent reaction substrate, to form microdroplets such that each droplet contains at most one magnetic bead. In this case, each droplet contains at most one exosome. Only exosomes expressing distinct diagnostic biomarkers (GPC-1) are connected with a fluorescent reaction enzyme, which catalyzes the fluorescence of the substrate in the droplet. Finally, the digital detection of exosomes is realized by counting the fluorescent droplets. Zhao et al. [127] designed an ExoSearch exosome separation chip device to separate plasma exosomes from patients with ovarian cancer. The collected exosomes were stained with a fluorescent dye such as Cy-5, Rhodamine, and fluorescein isothiocyanate (FITC) labeling antibodies (anti-CA-125, anti-EpCAM, and anti-CD24) for multicolor fluorescence imaging at a low limit of detection of 7.5 × 10^5^ particles mL^−1^. The ExoSearch chip possesses two fluid injection ports on the chip: one end is used to inject the plasma sample to be tested, and the other end is used to inject the immunomagnetic bead solution containing specific antibodies. Through the serpentine channel, the Dean vortex and the inertial lift are generated to promote the complete mixing of the two streams and then captured by the magnetic field. Kanwar et al. [128] outlined an ExoChip chip with a layer of biotinylated CD63 antibody on the surface of the microfluidic device channel to capture exosomes containing CD63 antigens from the cell culture supernatant and serum of pancreatic cancer patients. The captured exosomes were quantified with a lipophilic membrane fluorescent carbocyanine dye (Dio).

### 4.4. Surface-Enhanced Raman Scattering Technology

Raman spectroscopy is a method to analyze the vibration mode of a sample by measuring the inelastic scattering effect caused by the radiation laser [129]. The combination of Raman spectra and microfluidic technology creates new opportunities for system miniaturization and integration [130]. Furthermore, SERS is a spectroscopic phenomenon that enhances the Raman signal by absorbing the molecule on the rough surface or nanometal materials [103]. In recent times, SERS has been employed to detect and differentiate exosomes derived from different cells [131,132,133]. The cancer-specific molecular composition differentiates the cancer exosomes from other exosomes [134]. Wang et al. [135] developed a microfluidic Raman biochip to isolate and analyze exosomes in situ. The exosomes were enriched by anti-CD63 magnetic nanoparticles through the mixing channel of the staggered triangular column array. The enriched exosomes were then magnetically fixed on the Raman spectra detection area, and antibody EpCAM functionalized Raman beads with high-density nitrile and used them as probes. By simply monitoring the intensity of the 2230 cm^−1^ SERS peak of prostate cancer and normal prostate cells, exosomes can be quantitatively detected. This novel device may have the potential as a clinical exosomal analysis tool for prostate cancer. Park et al. [133] further demonstrated the Raman spectra of lung cancer and normal cell-derived exosomes by combining SERS and statistical pattern analysis. In short, SERS spectra of exosomes were analyzed by principal component analysis. By employing this pattern analysis, lung-cancer-cell-derived exosomes were clearly distinguished from normal ones with a sensitivity of 95.3% and a specificity of 97.3%. Shin et al. [136] demonstrated an accurate diagnosis of early-stage lung cancer using exosomal deep learning-based SERS. The deep learning model predicted that plasma exosomes of 90.7% of patients (*n* = 43) had higher similarity to lung cancer cell exosomes than the average of the healthy controls and lung cancer with an area under the curve (AUC) of 0.912 for the whole cohort, and stage I patients with an AUC of 0.910. These results exposed the tremendous potential of the combination of exosomal analysis and deep learning as a method for early-stage liquid biopsy of lung cancer.

### 4.5. Aptamer-Based Separation Method

An aptamer with high specificity and affinity discerns and connects to their targets, such as antibodies, and has been used to establish affinity-based isolation of exosomes [137]. For instance, a coating agent composed of EpCAM-affinity peptide aptamer (Ep114) and zwitterionic poly-2-methacryloyloxyethyl phosphorylcholine (MPC) polymer has been used for exosome isolation [138]. Kaushik et al. [139] took advantage of MB@SiO2@Au nanoparticles modified with a CD63 nucleic acid aptamer to capture exosomes from cancer patients’ plasma. In order to achieve high efficiency, sensitive and specific detection, Darmanis et al. [140] developed a sensitive plasma protein analysis by microparticle-based proximity ligation assays, which relies on the simultaneous recognition of the target protein by three antibody molecules to enhance specificity. This method can perform stable and highly sensitive protein detection in complex biological materials based on paramagnetic particles. Wan et al. [141] construed a method to modify DNA nanocomponents depending on molecular recognition on the surface of nano-scale exosomes. This in situ assembly method is based on molecular recognition between DNA aptamers and their exosomal surface biomarkers, and DNA hybridization chain reactions induced by aptamer-chimeric triggers. It is worth noting that functional DNA nanostructures are assembled on nano-sized exosomes. This strategy has laid a scientific foundation for further exploration of biomedical applications related to exosomes.

Compared with antibody-based capture methods, aptamer-based capture methods may have a higher potential in exosomal separation because of their high binding affinity toward the protein biomarker on the surface of tumor-derived exosomes [142].

### 4.6. Quantum Dot-Based Exosome Quantification

In many electrochemical sensors, a signal amplification step based on nanomaterials is usually used to detect immune response events [143,144]. In particular, incorporating inorganic-colloid tracers such as quantum dots (QDs) as a signal transduction label, combined with an anodic stripping voltammetric readout, was widely reported to significantly enhance the sensitivity of immunoassays [145,146].

To date, several other electrochemical methods based on aptamer and horseradish peroxidase (HRP) mediated amplifications have also been developed and executed to quantify exosomes [147,148]. For example, Wu et al. [149] developed sensors based on magnetic and fluorescent biological probes (MFBPs) for one-step quantification of exosomes in oral cancer. Within the MFBPs, self-assembled DNA concatamers loaded with numerous QDs were ingeniously tethered to aptamers and anchored on the surface of magnetic microspheres. The aptamer’s efficient recognition and capture of an exosome would simultaneously provoke the release of a DNA concatamer as the detection signal carrier, thereby generating a “one exosome-numerous QDs” amplification effect. This research provides a new, universal strategy for sensitive and quantitative analysis of exosomes from body fluids, thereby promoting the development of exosome-based liquid biopsy techniques. Moreover, Boriachek et al. [107] combined exosomes on magnetic beads with exosomal-specific antibodies, then used CdSeQD-functionalized specific antibodies to isolate tumor-derived specific exosomes. Here, QDs are used as signal amplifiers to determine tumor-derived specific exosomes by combining voltammetry and immunological technologies. This method is predominantly applied to tumor-derived specific exosomal protein antibodies such as FAM134B for colon and human epidermal growth factor receptor-2 (HER2) for breast cancer, which represents a promising bioassay technology.

### 4.7. Emerging Nanomaterial Detection Technology

The covalent organic framework (COF) has received more and more attention due to its diverse structure and distinct functions. Wang et al. [150] researched, designed, and manufactured a new type of COF-based nanoprobe, in which spherical COFs are functionalized with gold nanoparticles (AuNPs) modified with para-sulfocalix[4]arene hydrate (pSC4) and HRP, called HRP-pSC(4)-AuNPs@COFs. It was subsequently used for the electrochemical detection of colorectal cancer-derived exosomes. In this design, pSC(4), as a friendly linker, can recognize and bind various amino acid residues on the surface of exosomes, whereas AuNPs, with excellent conductivity, can accelerate the migration of charge carriers and improve the response of the biosensor. It is worth noting that the high porosity of COF allows them to load a large amount of HRP, giving colorectal cancer a high catalytic activity. This method has also been used to analyze clinical serum samples and can successfully distinguish colorectal cancer patients from healthy ones, indicating considerable potential in clinical diagnosis. Moreover, Wang et al. [151] designed a microfluidic device with a multi-dimensional layered structure of silicon cilia micropillars, which is primarily based on the size of the particles to achieve exosomal separation and capture exosomes with a diameter of 30~200 nm. The device electrolessly etches silicon nanowires on the sidewalls of micropillars with the aid of electrodeposited silver nanoparticle catalysts. A porous microstructure is then formed to capture particles, and finally, the porous silicon nanowires are dissolved in phosphate buffer (PBS) to acquire exosomes. The nanowire has a long and thin morphology and a large surface area, which facilitates modification of the surface and assists in the capture of the target particles. Lim et al. [152] designed a magnetic nanowire device to capture exosomes. The surface of the nanowire is covered with three kinds of antibodies: CD9, CD63, and CD81, which can capture exosomes expressing these three membrane proteins in the sample. Iron oxide nanoparticles can be captured by a magnetic field, and finally, dithiothreitol is processed to obtain exosomes.

## 5. Single Exosome Phenotyping Technique

The phenotypic analysis of a single exosome is a new technology that perfectly combines immunology and optics [153]. With the help of single exosome phenotyping technology, oncologists can analyze tumor-related exosomes in more depth so as to better conduct tumor pathophysiological research, early screening, diagnosis, and clinical management. Hoshino et al. [154] determined the specific protein biomarkers derived from cancer cells by detecting the exosomal proteome of tumors, tumors’ surrounding tissues, and serum. After further analysis of the exosomal proteome in cancer patients, healthy tissues, and serum, the specificity and accuracy of the test results attained more than 90%. Clearly, by detecting the contents of serum exosomes, the type of tumor can be determined. The researchers used a single exosome phenotyping technique to analyze exosomes derived from human plasma and successfully detected exosomes carrying moesin and β_2_-microglobulin proteins, and quantitative analysis was performed on them. Crescitelli et al. [155] developed a method for isolating exosomes in order to distinguish the subtypes of exosomes derived from metabolic melanoma and analyzed the isolated exosomal subtypes. Next, the researchers divided the exosomes derived from melanoma tissues at different centrifugal speeds into two types: large and small. For the two exosomes, the single exosome phenotyping technique and ELISA method identified three protein biomarkers: CD9, CD63, and CD81. The single exosomes phenotyping technique also detected CD41a, a biomarker derived from platelets, which inferred that the sample was contaminated with trace amounts of exosomes derived from blood. Quaglia et al. [156] established that small exosomes of castration-resistant prostate cancer cells overexpress αVβ3 and can promote tumor growth and neuroendocrine differentiation. In contrast, small exosomes of cells that do not express αVβ3 do not have such an effect. Afterward, the researchers used a single exosome phenotyping technique to detect whether the isolated small exosomes contained αVβ3. The results projected that exosomes captured by the corresponding antibodies for protein biomarkers capable of forming ligands with αVβ3 (except CD41) contained αVβ3. However, for CD41, which does not form a ligand with αVβ3, αVβ3-containing exosomes were not detected. Härkönen et al. [157] used gastric cancer cells MKN74 to study the impact of CD44 expression on exosomal secretion, hyaluronic acid synthesis, and tumor cell growth. The researchers used a single exosome phenotyping technique to detect the content of CD44 and other biomarkers in the unpurified medium of MKN74 cells and the isolated and purified exosomes. The results exhibited that the MOCK (CD44 knockout) cell group had a significant decrease in the number of secreted exosomes in the unpurified medium and after separation and purification, compared with the control group. The above results signify that the single exosome phenotyping technique can directly detect the culture medium in a complex environment without the separation and purification steps, and the detection results are highly specific.

In the above studies, the single exosome phenotyping technique accurately detected the content of specific exosomes in body fluids and culture media with extremely high sensitivity and specificity. As an early diagnostic tool for cancer, exosomes and the application of exosome-loaded treatment methods will have a clear direction for cancer diagnosis and treatment research in the future. The single exosome phenotyping technology will provide accurate experimental results in future research to help humans overcome cancer.

## 6. Conclusions

Herein, various detection techniques of tumor-derived exosomes were systematically reviewed, and the clinical diagnostic conclusions based on exosomal molecular biomarkers and common cancers have been illustrated. Several portable, fully integrated exosomal biosensors provide a swift response, low detection limit, and high sensitivity, which can be plausible approaches for point-of-care testing.

As a minimally invasive method, exosomal research can reflect the overall molecular information of tumors and can be repeatedly sampled for large-scale monitoring. It has substantial advantages for tumor diagnosis and treatment. As natural vesicles of lipid bilayers, exosomes are also promising drug carriers and have tremendous potential for development in tumor treatment, providing research strategies for the individualized, targeted therapy of tumors. Due to their unique structure and function, exosomes have prospective applications under both pathological and physiological conditions. However, there are still certain limitations in the practical application of exosomes. Because many normal cells continuously release exosomes, it is incredibly challenging to isolate and analyze tumor-derived exosomes in a large population. As the original intent of liquid biopsy, qualitative detection of total exosomes may never achieve the goal of dynamically monitoring tumor progression. In order to apply exosomal technologies to clinical diagnosis as soon as possible, it is imperative to eliminate the interference of normal exosomes as far as possible, focus on the detection of tumor-derived exosomes subgroups, and discover more specific biomarkers for tumor exosomes so that single exosome detection is always the direction of future development. After all, most studies on protein biomarkers are limited to membrane proteins, whereas protein biomarkers in exosomes have not yet been explored. Therefore, proteomics analysis may promote the discovery of protein biomarkers inside exosomes.

## Figures and Tables

**Figure 1 ijms-23-00868-f001:**
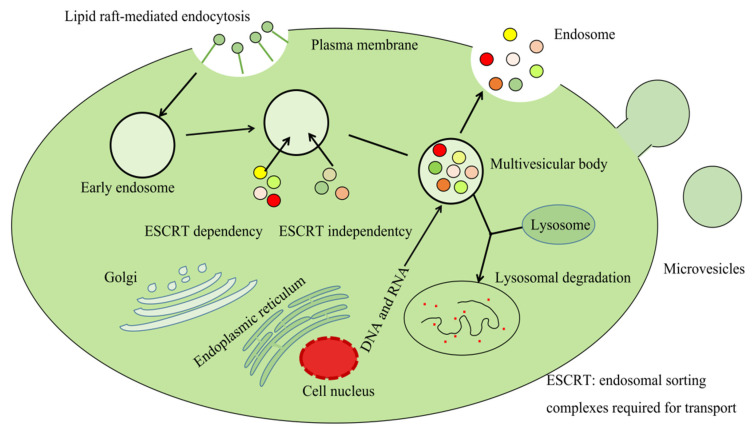
The schematic diagram of the synthesis and release of exosomes.

**Figure 2 ijms-23-00868-f002:**
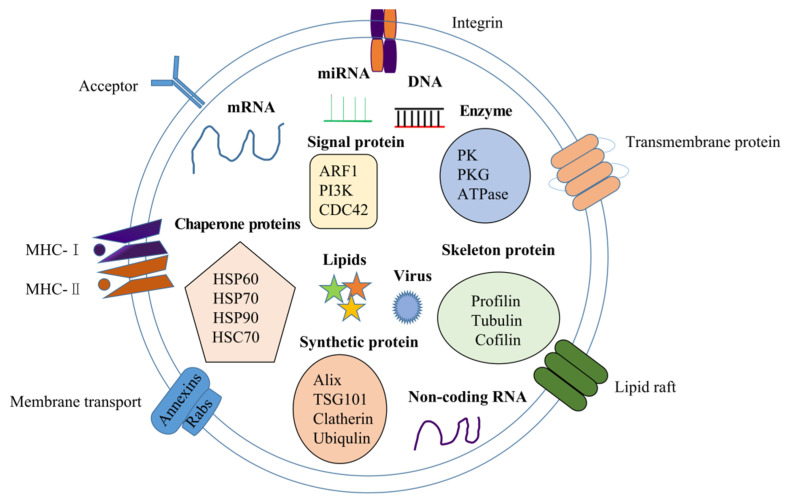
The schematic diagram of exosomal components.

**Figure 3 ijms-23-00868-f003:**
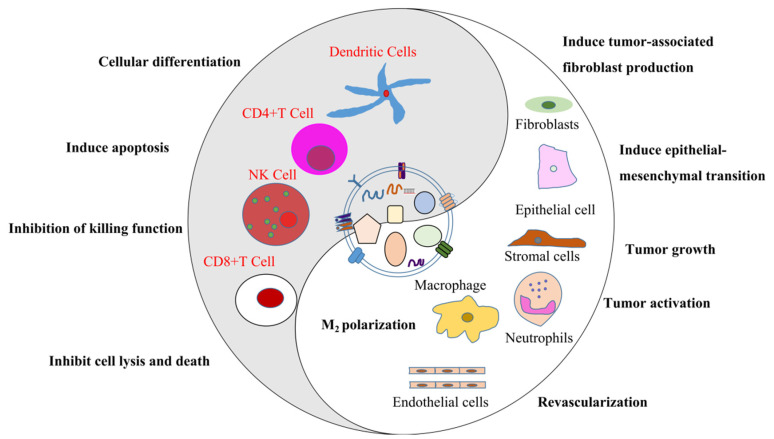
The schematic diagram of the relationship between tumor-related exosomes and tumors. Tumor-derived exosomes directly induce immune tolerance by transmitting inhibited or apoptotic signals to immune cells, indirectly affecting immune cells’ development, maturation, and anti-tumor activity. These exosomes alter the function of the responding cells by passing miRNA/mRNA/DNA to immune cells, polarizing macrophages into M2 type that accelerates tumor progression, converting normal fibroblasts into tumor-associated ones, promoting angiogenesis of endothelial cells, inhibiting the killing function of natural killer (NK) cells, and by triggering the differentiation of dendritic cells to promote the occurrence and development of tumors.

**Table 1 ijms-23-00868-t001:** Clinical significance of molecular markers of tumor-derived exosomes.

Scheme	Cancer Type	Marker Type	Name	Expression	Source	Clinical Value	Ref.
Central nervous	Glioma	DNA	IDH1 mutation	↑	Serum	Diagnosis, Prognosis	[35]
		mRNA	EGFRvIII		cerebrospinal fluid	Diagnosis	[43,44]
		miRNA	miR-454-3P/miR-320/miR-574-3P		Serum	Diagnosis, Prognosis judgment	[45,46]
		snRNA	RNU6-1		Serum	Diagnosis, Prognosis judgment	[46]
Thoracic	Lung cancer	mRNA	BRAF/EGFR/FRS2/GREB1/LZTS1	↑	saliva	Diagnosis	[47]
		miRNA	miR-451a/miR-425-3p/miR-4257		Serum plasma	Recurrence/Resistance/Prognosis judgment	[48,49,50]
		Protein	Tim-3/LBP/LRG1		Serum/plasma/Urine	Transferrer/Staging/Diagnosis	[51,52,53]
	Breast tumors	mRNA	hTERT		Serum	Early diagnosis/Recurrence	[54]
		miRNA	miR-223-3P		Plasma		[32]
		Protein	FN		Plasma		[55]
Digestive	Esophageal cancer	miRNA	miR-21/RUN6-1/miR-16-5p	↑	Serum	Diagnosis	[56,57]
	Gastric carcinoma	miRNA	miR-423-5p/miR-21,miR-1225-5p/miR-23b	↑/-/↓	Serum/Peritoneal lavage fluid/Plasma	Diagnosis, Prognosis judgment/Recurrence/Recurrence, Prognosis judgment	[31,58,59]
		LncRNA	UFC1	↑	Serum	Diagnosis, Prognosis judgment	[60]
		Protein	TRIM3	↓	Serum	Diagnosis	[61]
	Colorectal cancer	mRNA	KRTAP5/MAGEA3	↑	Serum	-	[62]
		LncRNA	BCAR4		Serum	-	
		miRNA	miR-6803-5P, miR-548c-5p, miR-92a-3p		Serum	Transferrer/Staging/Diagnosis	[63,64]
						Diagnosis/Transferrer	
		circRNA	Circ-KLDHC10		Serum	-	[65]
		Protein	TAG72/CA125/CPNE3		Plasma	Resistance/Transferrer/Diagnosis, Prognosis judgment	[66,67,68]
	Liver cancer	mRNA	hnRNPH1	↑	Serum	Diagnosis	[29]
		miRNA	miR-638	↓	Serum	Diagnosis	[69]
			miR-122	↑			[70]
			miR-148a	↑			
		LncRNA	LINC00635	↑	Serum	Diagnosis, Prognosis judgment	[71]
		Protein	LG3BP/PIGR	↑	Serum	Diagnosis	[72]
	Pancreatic carcinoma	miRNA	miR-191/miR-21/miR-451a	↑	Serum	Diagnosis	[73]
		Protein	GPC1^+^				[36]
Urinary	Bladder cancer	miRNA	miR-615-3p	↑	Urine	Diagnosis, Prognosis judgment	[30]
		LncRNA	MALAT1/PCAT-1		Urine		[74]
		Protein	TACSTD2/EDIL-3		Urine		[75,76]
	Kidney cancer	miRNA	miR-210/miR-1233		Serum		[77]
		Protein	MMP9/PODXL/DKK4		Urine		[78]

**Table 2 ijms-23-00868-t002:** Comparison of seven types of detection strategies for exosomes and tumors.

Methods	Mechanism	Cancer	Advantage	Limitation	Significance	Ref.
Size exclusion chromatography	Substances eluted out in accordance with their particle size or charge difference	Colon cancer Ovary cancer Liver cancer Astrocytic glioma	Swift preparation Keep native state of exosomes Adequate reproducibility Potential for both small and large sample capacity; Capable of processing all types of samples	Relatively high device costs Additional methods for exosomal enrichment are required Longer time	This method is not only suitable for processing trace amounts of liquid samples but also easily scalable and automated for high-throughput exosomal preparation, which allows fast, precise, scalable, and automated exosomal isolation.	[91,92,93,94]
Droplet digital enzyme-linked immunosorbent assay	Based on the specific binding between exosome biomarkers and immobilized antibodies (ligands)	Breast cancer	Ultra-sensitive detection Suitable for separating exosomes of specific origin; High-purity exosomes Absolute quantification No chemical contamination	High-cost antibodies; Exosomal biomarkers must be optimized Low processing volume and yields Extra step for exosomal elution may damage native exosome structure	Collecting exosomes of specific origin not only facilitates the study of their parental cells but also provides essential indicators for disease diagnosis (for example, via detecting EpCAM positive exosomes to assess the existence of EpCAM related cancers).	[95]
droplet digital PCR	ddPCR technology uses a combination of microfluidics and proprietary surfactant chemistries to divide PCR samples into water-in-oil droplets	Lung cancer	Reproducibility Precision Easier to set up, faster Higher sensitivity Does not require complex informatics support for analysis.	Necessitates the knowledge of genetic or epigenetic changes to be detected Limited multiplex abilities	ddPCR uses aqueous droplets with volumes ranging from a few femtoliters to nanoliters dispersed in oil for compartmentalization of PCR reactions, opening up the possibility of having a theoretically unlimited number of compartments, thus largely increasing detection sensitivity.	[96,97,98,99]
Microfluidic technology	Based on different principles, including immunoaffinity, size, and density	Ovarian cancer Breast cancer Pancreatic cancer	High-throughput Low material consumption Highly efficient Cost-effective Portable Easily automated and integrated with a diagnosis	Low sample capacity	Microfluidic techniques are dramatically innovating the landscape of exosome-based diagnosis by transferring the traditional two-step procedure (exosome isolation and characterization) to an integrated one-step process, which is especially valuable for non-invasive disease detection, such as early-stage cancer screening.	[100,101,102]
Surface-enhanced Raman scattering technology	SERS is a spectroscopic phenomenon that enhances the Raman signal by absorbing the molecule on the rough surface or nanometal materials.	Nonsmall cell lung cancer Pancreatic cancer	High enhancement factor Stable No quenching and photobleaching Fingerprint characteristics and narrow Raman bands	Raman spectra of exosomes are complex and nonconforming.	Multiple analytes (or biomarkers) in one sample could be detected in a single cycle/run when using more than one SERS tag, which results in an accurate, efficient, and simplified diagnosis of diseases.	[103,104,105]
Aptamer-based separation method	The aptamer is combined with oligonucleotides or peptides as a detection probe, linked to a biological vector or emerging nanomaterials to achieve target monitoring	Prostate cancer Lung cancer Breast cancer	Aptamers are selected for cell surface biomarkers in their native state, and conformation without previous knowledge of their biomarkers High specificity and affinity Aptamer-cell affinity interaction	Bind and release cells on-demand	The biomarkers and corresponding aptamers can be exploited to improve cancer diagnostics and therapies.	[106]
Quantum dot-based exosome quantification	Using quantum dots as signal amplifiers	Breast cancer Colon cancer	High specificity and sensitivity Using QDs for the signal enhancement Considerable reduction in biofouling issues	Magnetic wash Purification steps	The approach could potentially represent an effective bioassay for the quantification of disease-specific exosomes in clinical samples.	[107]

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
