# Peer review of "Emerging Advances of Detection Strategies for Tumor-Derived Exosomes"

_ijms, 2022, doi:10.3390/ijms23020868_

Round 1
Reviewer 1 Report
The submitted manuscript constitutes a comprehensive summary of the innovative research on numerous detection strategies for tumor-derived exosomes. The review in my opinion can be accepted after minor revision:
- what is the source of Fig1? if this was taken from the literature it should be clearly stated.
- the same for Fig 2 and Fig 3- information about the source should be placed in the description of the figure.
- Table 1 - please make the table more readable. Remove horizontal lines which are not necessary.
- Table 2 - maybe the Authors could re-arrange the Table to take it more readable?
- please check the recent publications one more time, e.g. https://doi.org/10.3390/cancers12113315
Reviewer 2 Report
The Review by Cheng et al. summarizes the recent exosome-discovery based approaches in cancer. The review is well done.
- The following reference is suggested to be included and commented in the section “tumor development”
doi:10.3390/ijms22168430
doi: 10.3390/cancers13112792
doi: 10.1038/s41389-020-00274-y
- Table 1 the n of “expression" is out of line
- Table 2 needs restyling. This version is confusing
